# Programming Effects of Prenatal Stress on Neurodevelopment—The Pitfall of Introducing a Self-Fulfilling Prophecy

**DOI:** 10.3390/ijerph16132301

**Published:** 2019-06-28

**Authors:** Laura S Bleker, Susanne R De Rooij, Tessa J Roseboom

**Affiliations:** 1Department of Obstetrics and Gynaecology, Amsterdam University Medical Centers, Meibergdreef 9, 1105 AZ Amsterdam, The Netherlands; 2Department of Clinical Epidemiology, Biostatistics and Bioinformatics, Amsterdam University Medical Centers, Meibergdreef 9, 1105 AZ Amsterdam, The Netherlands

**Keywords:** prenatal stress, fetal programming, neurodevelopment

## Abstract

There is increasing interest for the potential harmful effects of prenatal stress on the developing fetal brain, both in scientific literature and in public press. Results from animal studies suggest that gestational stress leads to an altered offspring neurodevelopment with adverse behavioral and cognitive consequences. Furthermore, there are indications in human studies that severe prenatal stress has negative consequences for the child’s neurodevelopment. However, stress is an umbrella term and studies of maternal stress have focused on a wide range of stress inducing situations, ranging from daily hassles to traumatic stress after bereavement or a natural disaster. Mild to moderate stress, experienced by many women during their pregnancy, has not consistently been shown to exert substantial negative effects on the child’s neurodevelopment. Additionally, the vast majority of human studies are observational cohort studies that are hampered by their fundamental inability to show a causal relationship. Furthermore, our limited knowledge on the possible underlying mechanisms and the effects of interventions for prenatal stress on child neurodevelopmental outcomes emphasize our incomplete understanding of the actual effects of prenatal stress on child neurodevelopment. Until we have a better understanding, it seems counterproductive to alarm all pregnant women for possible harmful effects of all sorts of prenatal stress, if only to avoid the induction of stress itself.

## 1. Prenatal Stress Research

An individual’s neurodevelopmental trajectory is not only determined by their genetic heritage, but also by interaction with the environment, in particular during sensitive periods of rapid cell division throughout fetal and early postnatal life. An adverse early environment may lead to neurodevelopmental changes that have lasting negative effects on health over the lifespan. During the last decades, there has been increasing scientific interest in studying adverse programming effects of the ‘environmental’ factor maternal psychosocial stress in pregnancy on offspring neurodevelopment. A non-systematic search in PubMed using the terms maternal stress and child/offspring development, crudely reflecting all articles that have been published on programming effects of prenatal stress, yielded a total of 3824 articles, of which the majority (62.8%) were published in the past decade (Figure 1). 

## 2. Evidence from Animal Studies

Animal models in which the effects of stress during pregnancy on offspring neurodevelopment were investigated, have demonstrated that applying various types of stress during gestation leads to anxiety- and depression-like behavioral patterns, impaired learning and attention deficits, and reduced social interaction in offspring [1,2]. However, the type of stress applied in animal studies, typically in rodents, varies widely and also differs from the multifaceted phenomenon of stress as it is experienced by human beings [3]. Repeated restraint, social isolation, immobilization, noise exposure, strobe lighting, electric shocks, hypoxia, forced swim, exposure to cold, food deprivation, and administration of dexamethasone injections are some examples of stressful paradigms that have been applied in animal models [2,4]. The specific effects of the different forms of gestational stress on various types of behavior in the offspring are not entirely clear, since usually, only one type is included in an experimental model. In addition, the strain of animals, the gestational age at which the stressor is applied, and the age of the offspring during the behavioral or cognitive tests differs widely between studies. Despite this heterogeneity in study methodology, the evidence is convergent in that in animals prenatal stress consistently leads to higher stress responses, more anxiety and depression-like behavior, and reduced cognitive abilities in the offspring [1,2,4]. 

## 3. Evidence from Human Studies

Translating the findings from animal studies to human populations is challenging. There is evidence that transient fluctuations in early experiences such as stress could have greater long-term impacts in small, short-lived species compared with large, long-lived species such as humans. This implies greater buffering of the negative effects of early-life stress in humans compared to animals [5]. Furthermore, human studies on prenatal stress rely on observations of a wide variety of stressful experiences, traumatic events and mood disorders, which are commonly clustered under the term stress both in the scientific literature as well as in public press. Stress may refer to common daily hassles (e.g., work- or parenting-related), but also to severe traumatic experiences such as domestic violence or the death of a close relative. Another category is formed by the acute stress caused by natural disasters, terrorist attacks, or wars. Thus, stress is used as an umbrella term for a whole range of phenomena reflecting different types of negative experiences or feelings ranging from uncomfortable (mild) to extremely traumatic (severe). It seems unlikely that all these types of stress induce a similar biochemical response in the pregnant women and that they have similar consequences for the offspring. Mild to moderate stress is also highly common. For example, in healthy populations of pregnant women across the globe, mild to moderate stress is reported by more than half of respondents [6,7]. If this level of stress would lead to an impaired neurodevelopment of the unborn child following the fetal programming paradigm, the majority of children alive today would be affected. The number of women who experience more severe forms of stress during their pregnancy is substantially lower. For example, intimate partner violence (IPV) or post-traumatic stress disorder (PTSD) during pregnancy are reported by much smaller numbers of women, 3% and 3.3% for IPV and PTSD respectively [8,9]. ‘Internal’ causes of stress, such as psychopathology including a clinical depressive disorder or a general anxiety disorder affect respectively 11.9% [10] and 4.1–15% [11] of pregnant women. However, studies that use a screening tool to identify women at risk of developing a depressive or anxiety disorder report much higher numbers: up to 25% during pregnancy [11,12]

The substantial differences in percentages of pregnant women experiencing these various types of prenatal ‘stress’ reflects the unique nature of each stressor, in terms of its cause, symptoms and treatment. The potential consequences for child neurodevelopment of each distinctive type of stress should be distinguished accordingly, because they are likely to be very different. 

The conceptualization and quantification of ‘stress severity’ is extremely challenging, and cut-off scores for ‘severe’ symptoms of stress, depression, or anxiety vary among studies. Nevertheless, studies on associations between extreme prenatal stress, for example caused by the death of a loved one or a natural disaster, and child (neuro)development in humans generally suggest that severe prenatal stress is associated with an increased risk for preterm birth, low birth weight, and a wide variety of neurodevelopmental, behavioral, and emotional disorders as well as impaired cognitive function. For example, women who experienced the death or serious illness of a close relative during pregnancy or in the 6 months before conception gave birth to babies with an average weight of 27 grams lighter compared to babies born from women who did not experience such an event [13]. Another study showed that severe objective stress during pregnancy caused by an Ice Storm in the Canadian Provence of Quebec resulting in 40 days of power loss was associated with lower IQ scores of the offspring at age 5.5 years [14]. In contrast to these severe forms of stress, studies on milder forms of prenatal stress, such as elevated anxiety and depression symptoms without an actual clinical diagnosis, daily hassles, or parenting stress, do not consistently show an increased risk for poorer neurodevelopment in children [15,16]. For example, in one study, prenatal depression, anxiety, and daily stress symptoms were measured during the second trimester in financially stable, well-nourished women without traumatic experiences in pregnancy. Neither depression, nor anxiety, nor non-specific stress predicted poorer child development at age two, suggesting that in non-clinical populations, stress is not harmful for the developing child’s brain [17]. 

Another important consideration in interpreting human studies that describe associations between prenatal stress and child neurodevelopmental outcomes is their fundamental inability to prove a causal relationship. Some of these studies use a cross-sectional design, where women are asked to recall their pregnancy and report whether they were stressed at the time. This is highly susceptible to recall bias, as mothers of children with behavioral deficits may be more prone to report prenatal stress in retrospect. Observational studies, in which women are prospectively followed from pregnancy onwards, eliminate recall bias. However, observational studies are hampered by the presence of co-occurring risk factors that may relate to both prenatal stress as well as an altered child developmental trajectory, such as socioeconomic status or smoking behavior. Additionally, postnatal alterations in maternal behavior likely explain at least part of the associations observed between prenatal stress and child neurodevelopment. This is reflected by the fact that in many studies associations between prenatal stress and an adverse neurodevelopmental outcome in children disappear after adjusting for postnatal maternal mood [15].

## 4. Consequences for Society

The growing body of literature on prenatal stress and child neurodevelopment has led to increased attention for prenatal stress and following recommendations to prevent stress during pregnancy [15,18,19,20]. Although these recommendations are aimed at preventing prenatal stress, they may paradoxically induce it. In the non-pregnant population, it has been shown that the perception that stress impacts health is associated with poor (mental) health [21]. The increased awareness about stress among women of childbearing age was reflected in a poll we recently conducted on an online platform (www.womb-project.nl) among platform visitors (mostly women between 18–40 years of age). In this poll, stress during pregnancy was chosen by 32% of the voting women of reproductive age, above other topics including smoking, nutrition, sleep, exercise, and infant attachment as a topic related to child development that “requires more attention from researchers”. Although we do not know the exact reasons why the majority of women of childbearing age chose stress as the most important topic to be studied, the poll results indicate that the topic highly appeals to women of childbearing age. As a scientific community, we should consider whether these apparent concerns are well-grounded, in particular since the majority of pregnant women is likely to experience some sort of stress. Distressing the majority of pregnant women by communicating to them that stress will directly harm their unborn child seems intuitively counterproductive. Warning women about the harmful effects of stress may merely induce more stress. For example, a study including 3000 predominantly highly-educated white pregnant women measured psychosocial hassles, such as money worries, worries about the pregnancy, job problems, or fights with the partner. It appeared that 64% of women could be classified as experiencing medium to high stress. High prenatal stress (reported by 27% of participants) was a significant predictor of maternal reporting of gastrointestinal illness, respiratory illness, and total illness of the child during the first year, and also to more urgent care visits and emergency department visits concerning the child [7]. The general public could draw the immediate conclusion that worries about money or a fight with a partner during pregnancy leads to a child with fragile health. However, an important limitation in this study, acknowledged by the authors in the discussion section of this article, is that the questionnaires on child illnesses were completed by the mothers. It is not inconceivable that women who report a lot of stress during their pregnancy are also more concerned later on about the health of their children and are thus more likely to seek medical aid compared to women who do not experience high stress. Moreover, more objective disease parameters, such as fever or actual hospitalization, were not increased in the children of mothers with high prenatal stress, indicating that in this study high prenatal stress did not seem to affect child disease risk. Such nuances are of great importance for the interpretation of study results and the subsequent recommendations that reach healthy pregnant women with occasional stress [7]. A final but no less important argument for reassuring pregnant women and eventually society about the effects of mild stress during pregnancy on child neurodevelopment is to alleviate the blame that is often placed on pregnant women themselves by society. Typically, pregnant women are publicly blamed for any type of behavior or experience that may adversely affect the health of the fetus, regardless of the significance of the actual impact, while the substantial contribution from her partner, her direct social environment, and society are unjustly being underexposed [22]. In fact, the role of society in the context of individual prenatal stress is substantial, and potentially modifiable. Chronic stress, depression, and anxiety in the prenatal period are consistently more frequent in low- and middle-income countries (LMIC) [23,24]. Poverty, war, and domestic and/or sexual violence, which occur more often in LMIC, all contribute to the increased levels of stress, depression, and anxiety described in pregnant women [25,26]. Hence, there is a huge potential for improving maternal mental well-being globally by addressing these risk factors individually, which is pre-eminently a role for local governments and global health organizations. 

## 5. Mechanisms 

An additional reason to apply caution in warning pregnant women for the dangers of prenatal stress for their child’s neurodevelopment is the lack of understanding we currently have about a clear mechanism providing a satisfactory explanation on how prenatal stress may affect the fetal brain in humans. In preclinical studies, we can directly study the biological alterations on a cellular level in prenatally stressed animals. These studies have shown that various types of gestational stress and excess maternal and fetal plasma corticosterone levels can induce epigenetic changes, leading to a reduced glucocorticoid and mineralocorticoid receptor density in the hippocampus, accompanied by a greater and prolonged hypothalamic-pituitary-adrenal (HPA) response to stress in the offspring [4,27]. Directly examining brain tissue is not feasible in humans, and studies that indirectly assess the assumed biological mechanisms, for example by studying buccal cells, cord blood, or placental tissue, are still in its infancy [19]. Increased HPA axis activity in the stressed pregnant woman resulting in higher levels of the human stress hormone cortisol has often been proposed as a key mediator linking maternal stress to an altered fetal neurodevelopment. However, studies in overall healthy pregnant women with low to moderate levels of stress have not shown robust support for the hypothesis that prenatal stress directly increases maternal cortisol levels [28,29]. On the other hand, a study that included pregnant women with a clinically significant major depressive disorder did show a substantially altered cortisol dysregulation compared to healthy pregnant individuals [30]. This suggests that only in pregnant women with severe symptoms of ‘stress’, a substantial biological response in terms of heightened cortisol may be present. In line with findings from animal studies, depression and anxiety, as well as partner violence and war-related stress during pregnancy, have been associated with increased Deoxyribonucleic acid (DNA) methylation of fetal genes involved in the stress response in human studies [31]. However, DNA methylation studies in humans generally include small numbers. Additionally, in these studies, the alterations of DNA methylation have not yet been linked to long-term neurobehavioral changes in children, and the clinical relevance of changes in DNA methylation remains uncertain. Robust evidence from Randomized Controlled Trials that aim to investigate whether reducing prenatal stress by psychological treatments prevents epigenetic alterations, structural and functional brain measures and neurodevelopmental, behavioral, emotional, and cognitive deficits should provide further insight. Such experimental trials in humans pose many methodological challenges and have only recently started to emerge. Some small studies have provided clues that treatment of clinically depressed pregnant women improves short-term infant outcomes and ameliorates DNA methylation of the glucocorticoid receptor gene in buccal cells [32,33]. However, these studies are underpowered, and substantially larger studies are warranted before we can conclude that prenatal psychological treatment of depression and anxiety has a beneficial effect on child neurodevelopment, and which biochemical alterations in the maternal-fetal compartment are involved. Intervention studies directed at other stressors such as daily hassles, or bereavement induced by the death of a relative, and how this relates to the neurodevelopment of the child needs to be further investigated, because as each different type of stressor may initiate a different biological cascade with unique effects on child neurodevelopment.

## 6. Involving Participants

Another strategy in research on prenatal stress and child neurodevelopment that has thus far been largely unexploited is a more active involvement of the target group during the design of prenatal stress studies. By exploring the beliefs, concerns, expectations, and needs in women who are pregnant or trying to conceive, we might be able to better align study outcomes relevant to women of childbearing age themselves as well as researchers and clinicians. There are many opportunities for patients to participate more actively in the entire research process, from identifying research priorities to disseminating and applying research findings [34]. A recent systematic review included 26 studies that examined the effect of patient and public involvement (PPI) on rates of enrolment and retention in clinical interventions. PPI modestly, but significantly, increased the odds of participant enrolment [35]. For studies on stress and maternal mental health this is particularly important, because women with severe symptoms often decline participation in clinical trials, partly explained by the stigma attached to mental illness [36]. The awareness of the importance of involving participants is growing, shown by the origins of the patient engagement movements, such as the Patient Engagement Collaborative (PEC) [37].

Eventually, we should aim to provide clear evidence-based advice to stressed pregnant women, and offer effective interventions. This may be of particular importance in socioeconomically disadvantaged populations and LMIC, in which poverty, housing insecurities, and war occur more often compared to high-income countries [23,38]. Furthermore, pregnant women who suffer from chronic high levels of stress, a psychiatric disorder, or severe stress caused for example by bereavement, should be identified and supported accordingly. Not only to help these women and prevent potential direct harm for the developing fetal brain, but also to address possible consequences of prenatal stress that can indirectly impair neurodevelopment of the child, for example poorer lifestyle choices [39]. 

## 7. Conclusions

In our opinion, stress research could improve by improving methodology and involving patients. We should acknowledge the many methodological challenges in prenatal stress research, which we have not yet been able to overcome. Therefore, it seems unsubstantiated, and even counterproductive, to alarm pregnant women with some degree of stress about possible harmful effects on their unborn child’s neurodevelopment, if only to prevent the prophecy from fulfilling itself. 

## Figures and Tables

**Figure 1 ijerph-16-02301-f001:**
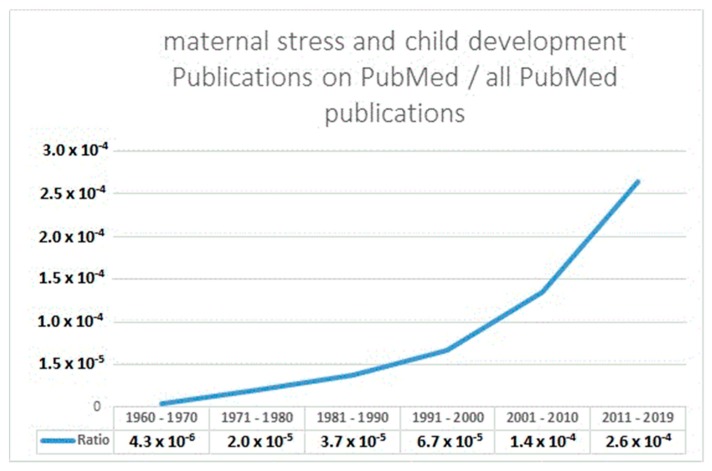
The relative portion of articles deposited on PubMed on maternal stress and child development.

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
