# Peer review of "Programming Effects of Prenatal Stress on Neurodevelopment—The Pitfall of Introducing a Self-Fulfilling Prophecy"

_ijerph, 2019, doi:10.3390/ijerph16132301_

Round 1
Reviewer 1 Report
This opinion paper addresses a very critical issue for pregnant mothers and their postnatal mental health related to the impact of physical and environmental stressors on the neurodevelopment of their offspring. The authors make a very convincing argument that the effects of various stressors on the fetus, neonate and newborn can be significant but the all stressors should not be treated or interpreted equally. Thus women that are pregnant should not be further stressed by being told that stress may be harmful to their child's neurodevelopment, with only a few exceptions (PTSD & IVP)
I have just a few suggestions about some sentences in the manuscript that should be slightly modified:
Line 65 and 66 on page 2 reads "response in pregnant and that". Either woman or mother should be added after pregnant.
Line 84 on page 3 reads "gave birth to babies of average 27 grams lighter". It should be revised to read "gave birth to babies with an average weight of 27 grams lighter"
Line 179 on page 5 could be amended to "child, needs to be further investigated".
Author Response
Response to Reviewer 1 Comments
This opinion paper addresses a very critical issue for pregnant mothers and their postnatal mental health related to the impact of physical and environmental stressors on the neurodevelopment of their offspring. The authors make a very convincing argument that the effects of various stressors on the fetus, neonate and newborn can be significant but the all stressors should not be treated or interpreted equally. Thus women that are pregnant should not be further stressed by being told that stress may be harmful to their child's neurodevelopment, with only a few exceptions (PTSD & IVP). I have just a few suggestions about some sentences in the manuscript that should be slightly modified:
Point 1: Line 65 and 66 on page 2 reads "response in pregnant and that". Either woman or mother should be added after pregnant.
Response 1: We would like to thank the reviewer for the critical evaluation of the manuscript, and for noticing the error, which we have adjusted accordingly (now line 72).
Point 2: Line 84 on page 3 reads "gave birth to babies of average 27 grams lighter". It should be revised to read "gave birth to babies with an average weight of 27 grams lighter”.
Response 2: We have changed the text as rightfully suggested by the reviewer (now line 97).
Point 3: Line 179 on page 5 could be amended to "child, needs to be further investigated".
Response 3: We have changed the text as rightfully suggested by the reviewer (now line 208 and 209).
Reviewer 2 Report
I agree with the authors that our understanding of the pathways linking maternal stress to offspring developmental outcomes is less than complete, and that emphasizing these links can induce their own stress in mothers. The basic message of tempering the certainty of these claims, and assuaging the fears of pregnant women, is sound advice, and a timely topic for an opinion piece in this journal. In general the piece is clear and easy to follow and I have only minor suggestions on how to strengthen this in revision.
The title and concept of a self-fulfilling prophecy is clever, but somewhat contradicts one major point made by the article – that there is little evidence that low level stress actually impacts the human fetus. If this is true, in what sense is the relatively low level stress associated with pregnancy specific anxiety (low level compared to bereavement, chronic stress etc) fulfilling the prophecy? I feel like the authors want it “both ways” – to discount the idea that we have evidence that low level stress is harmful, but to also claim that the low level stress induced by worrying about this is harmful. This is a minor issue – but the authors may want to ponder this as they finalize the framing in revision.
One point that is not made, but that I think should be (and I think will help make the overall point of the article), is the potential stigmatizing effects of these ideas, which can lead to mother-blaming, as emphasized elsewhere: Richardson, Sarah S., et al. "Society: don't blame the mothers." Nature News 512.7513 (2014): 131. That point seems important to make.
In a similar vein, I would have liked to have seen more acknowledgement that most forms of stress are structural and imposed by society. With rare exception (perhaps pursuing a highly competitive or otherwise stress-inducing career), women do not “choose” to be exposed to stress. And the women most at risk of exposure to stressors may have the fewest resources to overcome them. Think about inequality, neighborhood level effects (e.g. violence), lack of social capital, low income/demands of working without the option of maternity leave and so on. The solutions to such structural issues are typically going to be social/political rather than individual choices/action. This adds yet another wrinkle in emphasizing the intergenerational impacts of stress. This point just barely surfaces near the end of the piece, but it seems important to emphasize more centrally.
When reading the discussion of the lack of similarity between animal model stressors and human ones, I was struck that another point could be made to amplify this point. There is also evidence that the impact of a comparable maternal stressor scales with body size/life span of the species, with larger effects in small-bodied mice than in humans, and with intermediate effects in intermediate sized species (Kuzawa, C.W. and Thayer, Z.M., 2011. Timescales of human adaptation: the role of epigenetic processes. Epigenomics, 3(2), pp.221-234.). In other words, work on animals like mice not only relies upon often (by human standards) extreme stressors, but they also likely lead to an exaggerated sense of the possible effects that comparable stressors will have on human offspring. This is not an essential point to make, but I raise it in case the authors think it complementary to the points being made.
Author Response
Response to Reviewer 2 Comments
I agree with the authors that our understanding of the pathways linking maternal stress to offspring developmental outcomes is less than complete, and that emphasizing these links can induce their own stress in mothers. The basic message of tempering the certainty of these claims, and assuaging the fears of pregnant women, is sound advice, and a timely topic for an opinion piece in this journal. In general the piece is clear and easy to follow and I have only minor suggestions on how to strengthen this in revision.
Point 1: The title and concept of a self-fulfilling prophecy is clever, but somewhat contradicts one major point made by the article – that there is little evidence that low level stress actually impacts the human fetus. If this is true, in what sense is the relatively low level stress associated with pregnancy specific anxiety (low level compared to bereavement, chronic stress etc) fulfilling the prophecy? I feel like the authors want it “both ways” – to discount the idea that we have evidence that low level stress is harmful, but to also claim that the low level stress induced by worrying about this is harmful. This is a minor issue – but the authors may want to ponder this as they finalize the framing in revision.
Response 1: We thank the reviewer for critically evaluating our manuscript and for the suggestions given to improve the manuscript. We appreciate the point that is given by the reviewer on the manuscript’s title in relation to the message that we are aiming to communicate in the paper. The reason why we choose this title, is that we considered it a potential danger that in women who do not experience significant stress or some minor stress while they are pregnant, their pregnancy levels may actually increase to ‘toxic/chronic’ levels of stress (which might be harmful) when they believe that the stress that they (might) experience is invalidating their child’s health. If this would happen, then the prediction by the health care professionals or the media will come true indeed, even in cases of initial minor stress. We admit that there are many subtle nuances to add to this, which we have tried to address in the article, however, in the title we wanted to express our concerns about this subject in a way that is somewhat more ‘provocative’.
Point 2: One point that is not made, but that I think should be (and I think will help make the overall point of the article), is the potential stigmatizing effects of these ideas, which can lead to mother-blaming, as emphasized elsewhere: Richardson, Sarah S., et al. "Society: don't blame the mothers." Nature News 512.7513 (2014): 131. That point seems important to make.
Response 2: We fully agree with this statement, and have added this point in section 5 of the manuscript, to read: “A final but no less important argument for reassuring pregnant women and eventually society about the effects of mild stress during pregnancy on child neurodevelopment, is to alleviate the blame that is often placed on pregnant women themselves by society. Typically, pregnant women are publicly blamed for any type of behaviour or experience that may adversely affect the health of the fetus, regardless of the significance of the actual impact, while the substantial contribution from her partner, the social environment and society are unjustly being underexposed (ref).”, lines 157-163,
Point 3: In a similar vein, I would have liked to have seen more acknowledgement that most forms of stress are structural and imposed by society. With rare exception (perhaps pursuing a highly competitive or otherwise stress-inducing career), women do not “choose” to be exposed to stress. And the women most at risk of exposure to stressors may have the fewest resources to overcome them. Think about inequality, neighborhood level effects (e.g. violence), lack of social capital, low income/demands of working without the option of maternity leave and so on. The solutions to such structural issues are typically going to be social/political rather than individual choices/action. This adds yet another wrinkle in emphasizing the intergenerational impacts of stress. This point just barely surfaces near the end of the piece, but it seems important to emphasize more centrally.
Response 3: This is absolutely true, and we think this argument nicely follows the statement made in response to point 2 from the reviewer. We have extended the end of paragraph 4 to read: “In fact, the role of society in the context of individual prenatal stress is substantial, and potentially modifiable. Chronic stress, depression and anxiety in the prenatal period are consistently more frequent in low- and middle-income countries (LMIC) (refs). Poverty, war, and domestic and/or sexual violence that occur more often in LMIC all contribute to the increased levels of stress, depression and anxiety described in pregnant women (refs). Hence, there is a huge potential for improving maternal mental well-being globally by addressing these risk factors individually, which is pre-eminently a role for local governments and global health organizations.” Lines 163-169.
Point 4: When reading the discussion of the lack of similarity between animal model stressors and human ones, I was struck that another point could be made to amplify this point. There is also evidence that the impact of a comparable maternal stressor scales with body size/life span of the species, with larger effects in small-bodied mice than in humans, and with intermediate effects in intermediate sized species (Kuzawa, C.W. and Thayer, Z.M., 2011. Timescales of human adaptation: the role of epigenetic processes. Epigenomics, 3(2), pp.221-234.). In other words, work on animals like mice not only relies upon often (by human standards) extreme stressors, but they also likely lead to an exaggerated sense of the possible effects that comparable stressors will have on human offspring. This is not an essential point to make, but I raise it in case the authors think it complementary to the points being made.
Response 4: We thank the reviewer for suggesting this useful additional piece of evidence to strengthen our argument that animal and human studies are extremely difficult to compare. We think this statement fits nicely in the introduction in section 3, which now reads: “There is evidence that transient fluctuations in early experiences such as stress could have greater long-term impacts in small, short-lived species compared with large, long-lived species such as humans. This implies greater buffering of the negative effects of early-life stress in humans compared to animals (ref).” Line 60-63.
Reviewer 3 Report
The authors of this opinion piece focus on an important topic and the take-home message— that all stress is not bad for the child and we risk inducing maternal stress by sounding alarm bells about effects of stress— is one that deserves hearing.
A few comments to strengthen the impact of this piece.
Figure 1 could be misleading since the overall number of publications in Pubmed has also increased over the same time period. The figure should be calibrated as a percent of total articles as well. Related, and as the authors point out, “stress” is an umbrella category, so the search term “maternal stress” is unlikely to capture all publications or the full range of stressors.
As Figure 1 suggests, there are thousands of articles on maternal stress, yet the authors choose to highlight findings from only a selected few. While this is not an exhaustive review, it’s still good to know how or why the authors chose the examples that they did.
The conceptualization of milder versus severe stress isn’t always clear or consistent. Psychopathology (anxiety and depression) is included as mild stressors, though they can be chronic and severe, with significant biological changes. There is also insufficient attention paid to the differentiation between outside stressful events and internal stress (e.g., depression) where the stress could be a proxy for some other underlying vulnerability that is passed on from mother to child.
Finally, while the authors’ recommendations for involving participants in research is novel, this is glossed over with a single sentence and reference (number 25). Similarly, the authors refer to the negative impact of widespread warnings about curtailing stress, but this too only resorts to a single reference (ref 16). In each case the value of their arguments is mitigated. Overall the piece could have more immediate impact if the discussion is more integrated, stress more precisely defined for various social contexts, and the involvement of participants more elucidated.
I look forward to reviewing an updated version of the manuscript should the authors be invited to, and choose to, resubmit.
Author Response
Response to Reviewer 3 Comments
The authors of this opinion piece focus on an important topic and the take-home message— that all stress is not bad for the child and we risk inducing maternal stress by sounding alarm bells about effects of stress— is one that deserves hearing.
A few comments to strengthen the impact of this piece.
Point 1: Figure 1 could be misleading since the overall number of publications in Pubmed has also increased over the same time period. The figure should be calibrated as a percent of total articles as well. Related, and as the authors point out, “stress” is an umbrella category, so the search term “maternal stress” is unlikely to capture all publications or the full range of stressors.
Response 1: We are grateful to the reviewer for putting forward a number of useful suggestions and critical remarks to increase the quality our manuscript. The reviewer is completely right in his opinion that figure 1 should show relative numbers. We have adjusted the figure accordingly, showing a persistent trend of increased publications on maternal stress relative to all publications. We acknowledge that the search term “maternal stress“ is rather crude due to the diversity of terminology in stress research. Our aim was to illustrate the overall increase of publications on (related) topics of prenatal psychosocial stress, and we chose “maternal stress” as a term as is quite broad. We believe that broadening our search even further by adding more types of stressors will definitely increase the number of hits, but overall will show a similar trend. Nevertheless, we have added an accompanying sentence now to read: “A non-systematic search in PubMed using the terms maternal stress and child/offspring development, crudely reflecting all articles that have been published on programming effects of prenatal stress, yielded a total of 3824 articles, of which the majority (62.8%) were published in the past decade (Figure 1).” Line 35-38. We sincerely hope the reviewer is satisfied with our response.
Point 2: As Figure 1 suggests, there are thousands of articles on maternal stress, yet the authors choose to highlight findings from only a selected few. While this is not an exhaustive review, it’s still good to know how or why the authors chose the examples that they did
Response 2: We have selected a few articles that accurately reflect the various topics that we wanted to highlight in the paper. The articles that we have selected to discuss in more detail, echo, in our opinion, the current general direction of the trend that is seen in comparable studies (with similar stressors, population etc.). As the reviewer mentions, the complete body of literature on all the various forms of stress is huge, and therefore extends the aim of our paper. We acknowledge that an opinion piece in particular may be prone to selective bias, selecting those studies that support the view of the author. Evidently, there will be individual studies that propagate conclusions and opinions that are different to that of our own. To emphasize the fact that this piece reflects an opinion that could (or perhaps should) be discussed, we have added a sentence in the last paragraph of the article to read: “In our opinion, stress research could improve by improving methodology and involving patients..etc” (line 235-236).
Point 3: The conceptualization of milder versus severe stress isn’t always clear or consistent. Psychopathology (anxiety and depression) is included as mild stressors, though they can be chronic and severe, with significant biological changes. There is also insufficient attention paid to the differentiation between outside stressful events and internal stress (e.g., depression) where the stress could be a proxy for some other underlying vulnerability that is passed on from mother to child.
Response 3: We absolutely agree with the reviewer that the distinction between mild or severe (toxic) stress is per definition arbitrary, and may reflect completely other biochemical mechanisms that may exert programming effect by entirely unique pathways. It is also true that depression and anxiety can be either mild or severe, depending on the interpretation used by the authors of the separate studies. Studies using a clinical diagnosis as a criteria for depression and anxiety typically show more convincing evidence of some programming effects on the fetus, for example on the HPA axis, which we have pointed out in line 184-188. We do not attempt to pretend to be able to solve the problem of the many difficulties around conceptualizing stress as a concept in this article, but hope to have more clearly expressed the uncertainty around the definition of stress and the variety in potential consequences of stress exposure, by integrating the reviewers rightful suggestions throughout the paper in the relevant sections. Lines 80-84 :” ‘Internal’ causes of stress, such as psychopathology including a clinical depressive disorder or a general anxiety disorder affect respectively 11.9% [10] and 4.1–15% [11] of pregnant women. However, studies that use a screening tool to identify women at risk of developing a depressive or anxiety disorder, report much higher numbers, up to 25% during pregnancy [11, 12]”; lines 89-90: “The conceptualization and quantification of ‘stress severity’ is extremely challenging, and cut-off scores for ‘severe’ symptoms of stress, depression or anxiety vary among studies.”, and lines 205-208: “Intervention studies directed at other stressors such as daily hassles, or bereavement induced by the death of a relative, and how this relates to the neurodevelopment of the child, needs to be further investigated, because as each different type of stressor may initiate a different biological cascade with unique effects on child neurodevelopment”.
Point 4:
Finally, while the authors’ recommendations for involving participants in research is novel, this is glossed over with a single sentence and reference (number 25). Similarly, the authors refer to the negative impact of widespread warnings about curtailing stress, but this too only resorts to a single reference (ref 16). In each case the value of their arguments is mitigated. Overall the piece could have more immediate impact if the discussion is more integrated, stress more precisely defined for various social contexts, and the involvement of participants more elucidated. I look forward to reviewing an updated version of the manuscript should the authors be invited to, and choose to, resubmit.
Response 4: We have elaborated the section on involvement of participant involvement as suggested by the reviewer (line 216-223). As this topic is relatively novel, there is not much known yet about the effectiveness/barriers/methods of participant involvement in trials, but the evidence so far is promising. We have somewhat downgraded our statement on ‘widespread warnings’, and have rephrased this section to emphasize the increased attention on prenatal stress and the recommendation to prevent prenatal stress, as suggested by recent highly-cited reviews on the effects of prenatal stress of offspring. Also, we have cited a study showing that perception of stress as a negative influence on health is actually associated with poor (mental) health (in a non-pregnant population) (Keller, A., et al., Does the perception that stress affects health matter? The association with health and mortality. Health Psychol, 2012. 31(5): p. 677-84.), lines 126-127.